

# Young and new water fractions in soil and hillslope waters

Marius G. Floriancic[1,2], Scott T. Allen[3], James W. Kirchner[2,4,5]

[1] Dept. of Civil, Environmental and Geomatic Engineering ETH Zürich, Zürich, Switzerland
[2] Dept. of Environmental Systems Science, ETH Zürich, Zürich, Switzerland
[3] Dept. of Natural Resources & Environmental Science - University of Nevada, Reno
[4] Swiss Federal Research Institute WSL, Birmensdorf, Switzerland
[5] Dept. of Earth and Planetary Science, University of California, Berkeley, CA, USA

*Correspondence to*: Marius G. Floriancic (floriancic@ifu.baug.ethz.ch)

**Abstract.** The transport processes and corresponding times scales of water's infiltration into, and percolation through, the shallow subsurface are poorly understood. Here we characterize the transport of recent precipitation through a forested hillslope, using a continuous three-year record of O and H stable isotopes in precipitation, streamflow and soil waters from various depths. We found that the fractions of recent precipitation decreased with depth, both in waters extracted using suction-cup lysimeters and in waters extracted from bulk soil samples using cryogenic distillation. Not surprisingly, fractions
of recent precipitation found in soils and streamflow were much larger with wet antecedent conditions, showing that wet landscapes can transmit recent precipitation quicker than dry landscapes. Approximately 18% of streamflow was younger than 2-3 months, 11% was younger than three weeks and 7% was younger than one week; these new water fractions were similar to those seen in 20 to 80 cm deep soils. Mobile soil waters below 2 m depth contained much less recent precipitation (1.2±0.4% younger than two weeks) than streamflow did (12.3±2.1%), indicating that they are not the dominant source of
streamflow. Instead, streamflow must be generated from a mixture of deep subsurface waters, with very little isotopic seasonality and short-term variability, and shallow soil waters, with more pronounced isotopic seasonality and short-term variability. This study illustrates how flow, storage, and mixing processes linking precipitation to streamflow and evapotranspiration can be constrained by measuring isotopic variability across different hillslope positions, subsurface depths, and time scales.

## 1 Introduction

One third of Switzerland and 40% of the global ice-free landmasses (Waring and Running, 2007) are covered by forests. Thus, the modulation of input precipitation in forests is of great importance to the global freshwater cycle. While it is generally known that forest soils play an important role in the hydrological cycle by controlling infiltration and percolation to deeper storage (Sprenger et al., 2016), we have limited understanding of water movement in forested hillslopes, from
shallow subsurface storages to deeper storages or streamflow.




What we do know about subsurface water transport in forested hillslopes has been largely derived from stable isotopes and other tracers. For example, streamflow responds quickly to rainfall inputs, even though it may be mostly composed of old waters released from subsurface storage (i.e., the so-called "old water paradox"; Kirchner, 2003; Neal and Rosier, 1990).

More recently, hydrologists have recognized that water stored in the subsurface is often much older than the water draining from those same subsurface storages (Berghuijs and Kirchner, 2017; Kirchner *et al.*, 2023). Whereas most groundwater storages are dominated by waters with ages of 10 years or more (Jasechko *et al.* 2017), 25% of global streamflow is younger than 2 to 3 months (Jasechko *et al.*, 2016). While these apparent paradoxes are explainable (Berghuijs and Kirchner, 2017), we have few insights regarding where in the subsurface these age contrasts arise.


Water movement within shallower subsurface storages is often conceptualized as translatory flow (recent infiltration partly displacing and mixing with stored water; e.g.; Hewlett and Hibbert, 1967) or as preferential flow (by-passing the stored soil waters in the top layers; e.g.; Beven and Germann, 1982). Translatory flow would always result in more recent precipitation being closer to the surface, and preferential flow would result in recent precipitation also reaching deeper depths, potentially

without substantially mixing with waters stored in the intervening layers (i.e., bypass flow, a special case of preferential flow). Bypass flow is often argued to be responsible for ecohydrological separation (Brooks *et al.*, 2010), but it remains unclear how frequently bypass flow occurs. Thus, it also remains unclear how much waters flowing via macropores interact with waters within the soil matrix, especially under different soil wetness conditions; numerous previous studies found contrasting results regarding the degree of interaction between waters flowing through soils and waters stored in them (e.g.;

Geris *et al.*, 2015; Goldsmith *et al.*, 2012; Hervé-Fernández *et al.*, 2016). However, seasonal and event isotope signals often become more dampened with depth, indicating that the infiltrating water is becoming well mixed as it percolates to deeper depths (e.g., Sprenger *et al.*, 2019a; Barbecot *et al.*, 2018; Sprenger *et al.*, 2019b). While such phenomena have typically been investigated through sampling vertical profiles, it remains unclear how much these profiles are affected by lateral flow processes, which could also lead to damping of isotopic signals. The question that needs to be answered is, how do

subsurface transport processes yield the old, well-mixed waters often seen in streamflow?

To explore this question, we examine how precipitation events infiltrate into the subsurface, by analysing timeseries of stable water isotopes at various hillslope positions and depths. Soils typically carry the isotopic signature of many previous precipitation events. Due to seasonal isotopic signals in precipitation, with isotopically heavier precipitation in summer and

lighter precipitation in winter, we can track precipitation from different seasons in groundwaters (Jasechko, 2019; Jasechko *et al.*, 2014), streamflow (Allen *et al.*, 2019a) and soils (Sprenger *et al.*, 2019). We can use the seasonal fluctuations in precipitation to assess the relative proportions of younger and older water in streamflow. Specifically, the fraction of "young" water, defined as the fraction of streamflow that is younger than approximately 2-3 months, can be inferred from the amplitude of seasonal tracer cycles in precipitation and streamflow (Kirchner 2016a; 2016b). Alternatively, the fractions of

"new" water, defined as the fraction of water that is new since the last sampling, can be inferred from ensemble hydrograph





separation (Kirchner 2019; Knapp *et al.*, 2019; Kirchner and Knapp 2020) of tracer timeseries. Both methods have been widely applied to streamflow timeseries (e.g., Ceperley *et al.*, 2020; Jasechko *et al.*, 2016; Gentile *et al.*, 2023; von Freyberg *et al.*, 2017; von Freyberg *et al.*, 2018; Knapp *et al.*, 2019; Floriancic et al., 2023b) but few studies have applied them on timeseries of soil waters (Gallart *et al.*, 2020; Burt et al., 2023). Both of these stable isotope tools can reveal how

precipitation mixes with older storages as it travels down the soil profile and toward streams.

Here we use a 3-year continuous dataset of precipitation, mobile and bulk soil water, deep mobile water from boreholes, and streamflow to identify how the partitioning of young and new waters changes across different depths of a forested hillslope. We address the following research questions:


1) How much of streamflow and soil waters at different depths consist of young water (i.e., water that is younger than approximately 2-3 months) and new water (i.e., younger than 1 day to younger than 3 weeks, depending on the sampling interval)?

2)  To which extent do we find higher fractions of new water with wetter antecedent conditions?

3) Are the isotopic signals in subsurface waters variable along the hillslope and what do they reveal about streamflow generation?

## 2    Methods and available data

### 2.1 Assessment of young water fractions ($F_{yw}$) and new water fractions ($F_{new}$)

In seasonal climates the ratio of stable water isotopes ($^{18}O/^{16}O$ and $^{2}H/^{1}H$) in precipitation differs between summer and

winter, and also varies among individual precipitation events. Typically, precipitation in continental interiors is isotopically heavier in summer than in winter, resulting in a seasonal cycle of precipitation isotopes. Kirchner (2016a; 2016b) developed a method to calculate the so-called young water fraction $F_{yw}$, the fraction of streamflow that is younger than 2-3 months, from the ratio of amplitudes of the seasonal isotopic cycles in streamflow and precipitation. These amplitudes can be inferred from fitting sinusoids to the isotope timeseries, using an iteratively re-weighted least squares (IRLS) approach (see the R

script provided in the supplement of von Freyberg *et al.,* 2018). We calculated the seasonal cycle amplitudes for the precipitation, mobile soil water, bulk soil water, and streamflow timeseries, to estimate the young water fractions of water in soils and the stream.

In addition to young water fractions $F_{yw}$, we also calculated new water fractions $F_{new}$ via ensemble hydrograph separation as

described in Kirchner (2019). The major difference between $F_{new}$ and $F_{yw}$ is that whereas $F_{yw}$ estimates the fraction of streamflow or soil water that is younger than 2-3 months, $F_{new}$ estimates the average fraction of water that originates from precipitation between sequential pairs of sampling times (e.g., 3-4 days to 3 weeks for mobile soil waters, 2 to 3 weeks for



bulk soil waters and deep mobile soil waters, and 1 day to 3 weeks for streamflow). The method is based on correlations between the fluctuating isotopic signals in precipitation, soil waters, and streamflow. It reveals the average contribution from

an endmember (precipitation) to a mixture (soil water or streamflow) through correlations across multiple timesteps. This also makes it insensitive to unknown or unmeasured endmembers. While traditional hydrograph separation assesses how fractions of new and old water change over time (e.g., during an individual storm event) for each timestep, the ensemble hydrograph separation method (Kirchner, 2019) can estimate the average fractions of new and old water at different antecedent moistures and seasons. Scripts to perform this analysis in R and Matlab are available in the supplement of

Kirchner and Knapp (2020).

## 2.2 Sampling and data collection

All analyses are based on data collected in a forested hillslope-to-creek transect in a small 0.3 km² catchment (WaldLab Forest Experimental Site). This transect is dominated by spruce and beech trees and is part of the larger "Waldlabor Zürich" research and education initiative in Switzerland. This site, just north of Zurich, Switzerland, has a mean annual temperature

of 9.3°C and mean annual precipitation of 1134 mm (2010-2022). The soil is a luvisol of approximately 100 cm depth, on top of ~ 6 m of moraine material from the last glacial maximum. The dominant soil structure is silty sand, with clay fractions below 10%.

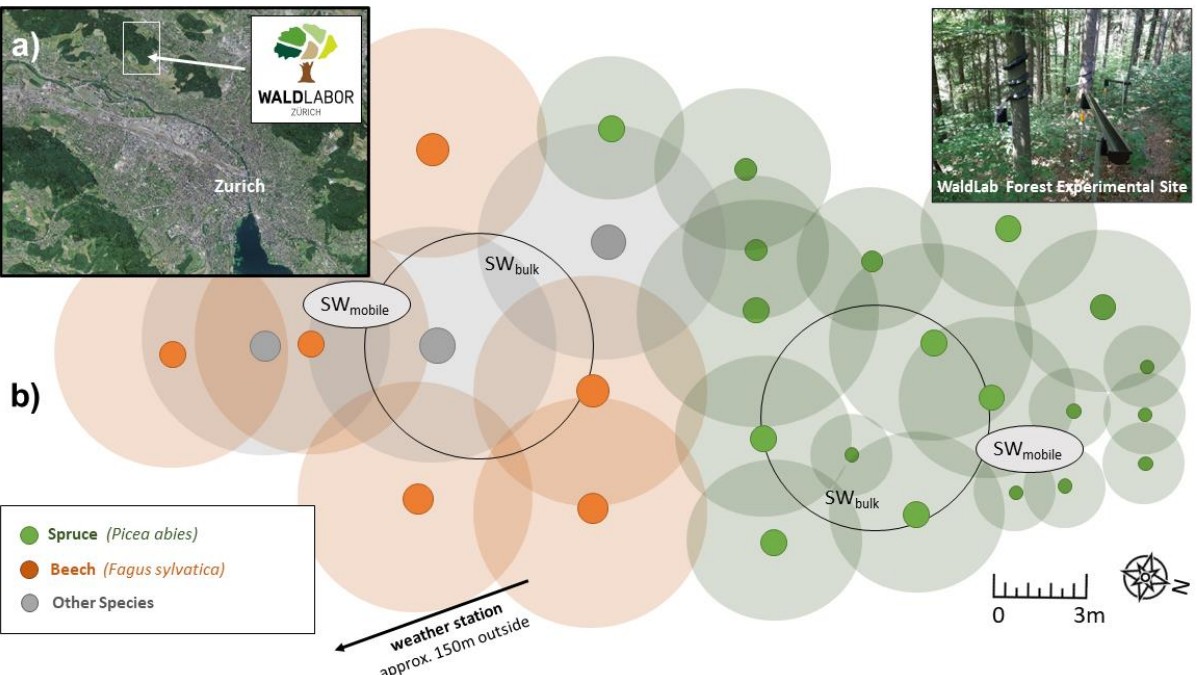

**Figure 1: Location of the "Waldlabor" in Zurich, Switzerland (a; Source: Swisstopo) and a schematic diagram of our WaldLab Forest Experimental Site (b), indicating the locations of trees (spruce, beech and other species shown in**




**green, orange and grey) as well as the locations of mobile ($SW_{mobile}$) and bulk ($SW_{bulk}$) soil water sampling. Precipitation for isotope analysis was sampled outside the forest perimeter at the weather station, at approximately 150 m distance from the site. The gauge of the "Holderbach" creek is located approximately 90 m from the**
**experimental site at the bottom of the hillslope.**

Since March 2020, we have measured major climate parameters outside the forest with a compact all-in-one weather station (Atmos 41 - Meter AG) at 10-minute resolution. Precipitation isotope samples were collected on any day when a precipitation event larger 3 mm occurred, using glass bottles with funnels and syringes to prevent evaporation and vapor
diffusion (as described in von Freyberg et al., 2020). Discharge was measured at a v-notch weir at the outlet of our experimental catchment with a pressure sensor (Keller AG – DCX-II) at 15-minute resolution. Daily streamflow samples at the outlet of the catchment were obtained with an ISCO 6712 autosampler (Teledyne Inc.) equipped with evaporation protection as described in von Freyberg *et al.* (2020), using a daily mixture of four 100 mL samples of streamwater taken every six hours (midnight, 6 AM, 12 PM, and 6 PM).


Soil water was sampled by multiple methods. Suction-cup lysimeters were used to sample what is commonly referred to as "mobile soil water" (the fraction of soil water that is held cohesively and can move freely) and bulk samples of soil were collected for water to be extracted by cryogenic distillation (assumed to comprise all water, including water in all capillary spaces). We sampled mobile soil water ($SW_{mobile}$) at 10, 20, 40 and 80 cm depths at two plots (Fig 1) with suction lysimeters
(Soil Moisture Equipment Corp., Slim Tube Soil Water Sampler). We applied a suction of 0.7 bar on Mondays and Thursdays and emptied the samplers twice a week on the following Thursdays and Mondays. In addition, we sampled bulk soil ($SW_{bulk}$) at the same two plots (Fig. 1) in 10, 20, 40 and 80 cm depths with a 2-cm-wide auger every three weeks, and extracted the bulk soil water cryogenically. Monitoring of mobile and bulk soil water at the shallower depths (10-40 cm) began in April 2020, and monitoring at 80 cm began in June 2021; thus the 80 cm records are significantly shorter.
Additionally, monitoring of a third "downslope" plot (not shown in Fig. 1) began in April 2022, specifically to investigate lateral flows after the overall study began.

Beyond those soil water collections, we also collected deep mobile waters from soil and the underlying moraine in 10 boreholes, screened between 2 m and 6 m depth, every two weeks. These boreholes were drilled in November 2020, and soil
/ sediment samples (for cryogenic water extraction) at different depths were collected into exetainers (Labco Ltd., 12 mL Exetainer,) and stored at -18°C for cryogenic extraction. Any water that collected in the bottom of these boreholes was sampled every two weeks, representing a seepage flux that is comparable to what can be collected by zero-tension lysimeters.



Cryogenic vacuum distillation was performed at the Institute of Agricultural Sciences Stable Isotope Lab at ETH Zurich (Grassland Science Group). The samples were evaporated in a water bath at a temperature of 80°C for three hours with a suction of $10^{-2}$ MPa, and the resulting vapor was trapped in u-shaped tubes immersed in liquid nitrogen (Sun *et al.*, 2022). We did not assess the extraction efficiency explicitly in this study, however in a previous study by Bernhard et al. (2023), we could show that extraction efficiencies for all samples exceeded 95 %. Extracted samples and all other samples

(precipitation, mobile soil water, deep mobile borehole water and streamwater) were stored in 1.5 mL glass vials (BGB Analytik) refrigerated at 2°C until analysis. The isotopic composition was analysed with a triple isotope water analyser (Los Gatos – TIWA-45-EP) with a precision of <1‰ for $^2$H and <0.2‰ for $^{18}$O, as determined by long-term replicate sampling of standards. All isotope data are reported in per mil (‰) notation relative to V-SMOW (Vienna Standard Mean Ocean Water) of $\delta^2$H; the respective $\delta^{18}$O plots can be found in the supplement.


Because the different types of samples were collected at different time intervals (i.e., after each event for precipitation, daily for streamflow, twice a week for mobile soil waters, every two weeks for deep mobile borehole waters and every three weeks for bulk soil waters), analyses involving comparisons of those samples to precipitation always used volume-weighted means of precipitation isotope values that were aggregated to reflect the same sampling intervals.

**3 Results and discussion**

**3.1 Seasonal signals in precipitation and streamflow**

The seasonal isotopic variation in streamflow was much smaller than that in precipitation, implying that the isotopic signal in precipitation was damped by storage and mixing on its way to becoming streamflow (shown for $\delta^2$H in Fig. 2 and for $\delta^{18}$O in the supplementary material Fig. S1). While precipitation isotopes did contain the expected typical seasonal cycle of lighter

isotopic signatures during the winter months and heavier isotopic signatures during the summer months, this seasonal cycle was much less pronounced in the streamflow isotope values. The median $\delta^2$H isotope values were -9.2 ‰ and -9.4 ‰ for precipitation and streamflow, respectively. Thus, overall streamflow was isotopically slightly lighter than precipitation, indicating that streamflow contained relatively more winter precipitation than summer precipitation.

From the ratio of seasonal amplitudes in streamflow and precipitation isotopes we estimated the fraction of young water (i.e., water that is younger than approximately 2-3 months, $F_{yw}$) in streamflow to be 18%. Thus, most of the streamflow at our site originated from water that is stored in the subsurface for longer than 2-3 months. This is not only true for our site but also in line with findings from across many other rivers across the globe (Jasechko *et al.*, 2016; von Freyberg *et al.*, 2018; Floriancic et al., 2023b). However, streamflow young water fractions by themselves cannot indicate where in the surface-to-stream

transit pathway this damping of seasonal cycles occurs.





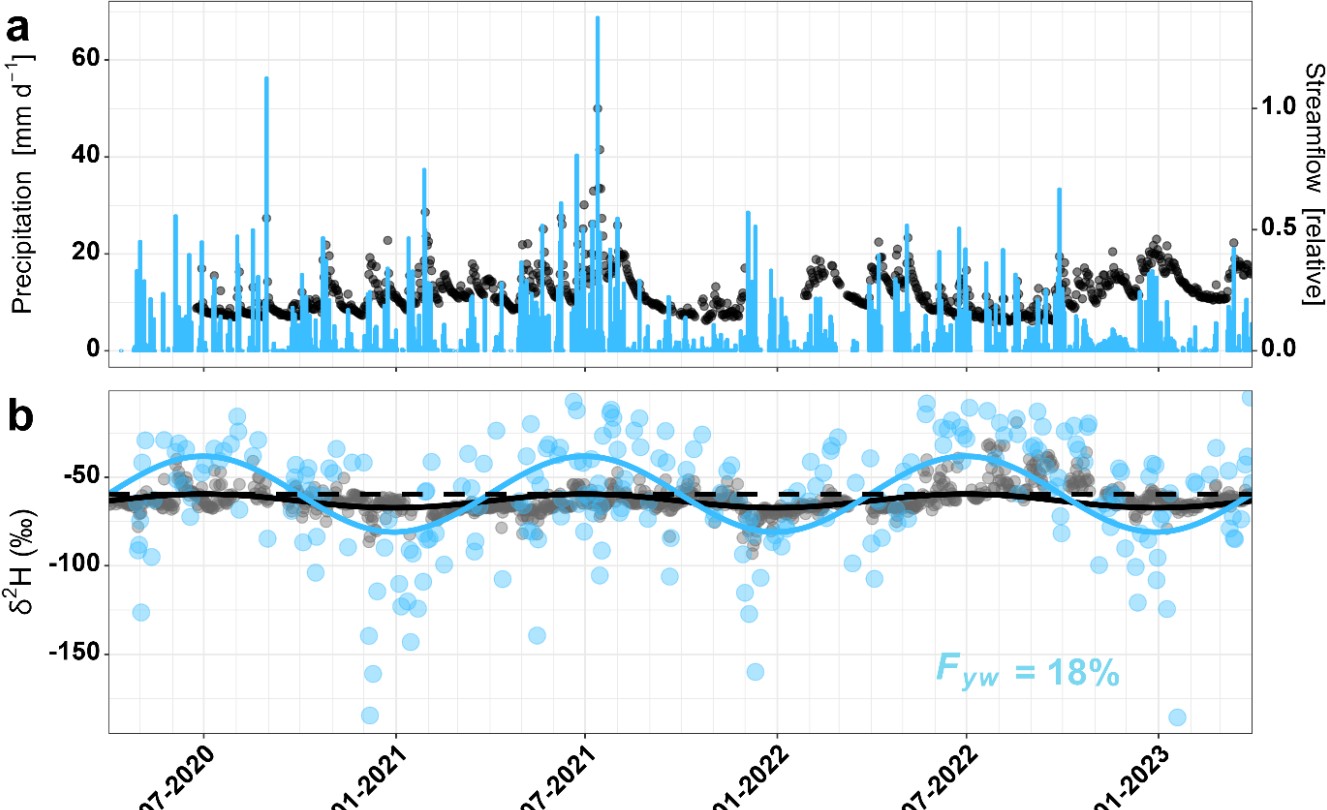

**Figure 2: Timeseries of precipitation (light blue) and streamflow (dark grey) (a) and their $\delta^2H$ isotopic compositions (b) from April 2020 until March 2023. Sinusoidal cycles were fitted to the isotope data using iteratively re-weighted least squares regression (in light blue for precipitation isotopes and in black for streamflow isotopes). The dashed black line indicates the volume weighed mean isotopic composition of precipitation; streamflow samples lying above and below this line indicate dominance by summer and winter precipitation, respectively. The seasonal cycles of the streamwater isotopes are damped relative to the precipitation isotopic cycles due to storage and mixing in the subsurface. The corresponding timeseries and sinusoidal fits for $\delta^{18}O$ can be found in Fig. S1.**

**3.2 Seasonal signals and young water fractions across different depths in the subsurface**

With increasing depth, observed sinusoidal cycles of soil water isotopes became increasingly damped relative to the precipitation input signal (Fig. 3). Values of $F_{yw}$ in mobile soil water – i.e., water sampled by suction-cup lysimeters – exhibited little variation in the top 40 cm (63% at 10 cm depth to 68% at 20 cm depth, and 63% at 40 cm depth), but decreased to 26% at 80 cm depth. As with the isotope ratios seen in precipitation across seasons, soil water at all depths showed heavier isotopes in summer and lighter isotopes in winter. The lack of a distinctive phase shift between precipitation





and soil water argues against a dominant role of piston flow, and supports the interpretation that recent (~2-3 month old precipitation) is seen in these soils, although at fractions that indicate significant dilution by older water (even, surprisingly, at shallow depths). Soil waters at 80 cm were heavily damped and typically lighter than average precipitation, indicating that recharge to these depths was overrepresented by winter precipitation.

Bulk soil $F_{yw}$ values were somewhat smaller than those of mobile soil water (Fig. 3). They decreased from 66% at 10 cm depth to 49% at 20 cm depth, 36% at 40 cm depth, and 18% at 80 cm depth, indicating greater dilution by old water with increasing depth (and a greater presence of old water in bulk soil than in mobile waters, at all depths > 10 cm). During all seasons, bulk soil waters were typically lighter than average precipitation, showing that winter precipitation predominates in these pore spaces, even in mid-summer. Suction-cup lysimeters disproportionately sample larger pore spaces that more readily fill and drain (Weihermüller *et al.*, 2005). Thus, while mobile waters in larger pores (sampled from suction-cup lysimeters) are filled by more recent precipitation, smaller pores in 20 to 80 cm depth are typically by-passed by recent precipitation and filled by waters predominantly originating from the winter season.

The seepage water collected from the boreholes at 2 and 6 m depths showed the least young water, with $F_{yw} = 6$ % (Fig. 3c). This is one third of the $F_{yw}$ observed in streamflow (~18%), implying that streamflow cannot be composed entirely of water from the deeper subsurface, but must also contain shallower components with larger seasonal isotopic cycles.



**Figure 3:** Timeseries of the $\delta^2$H isotopic composition from April 2020 until March 2023 in mobile (a) and bulk soil waters (b) of 10, 20, 40 and 80 cm depth and in deep mobile waters collected in boreholes of 2 to 6 m depth (c). Sinusoidal cycles were fitted to the isotope data using iteratively re-weighted least squares regression. The blue graph shows the sinusoidal cycle of precipitation. The dashed black line indicates the mean isotopic composition of precipitation, all samples above are dominated by summer precipitation, all samples below a dominated by winter precipitation. The seasonal cycles of soil waters exhibit increasing damping with depth. The timeseries and sinusoidal fits of the $\delta^{18}$O isotope signatures can be found in Fig. S2.





### 3.3 Recent precipitation in soil waters inferred by new water fractions

New water fractions ($F_{new}$) also decreased with increasing depth in the subsurface, with $F_{new}$ of the borehole deep mobile
water seepage being much smaller than that of streamflow (Fig. 4, Table 1). Mobile soil waters were sampled twice a week (albeit with significant gaps in the data during the Fall 2021 and 2022 dry periods), $F_{new}$ values calculated from these data reflect the fraction of soil water that is "new" on timescales of 3 to 4 days up to three weeks. This approach quantifies a much newer fraction of water than can be inferred from young water fractions (2-3 months).

The fraction of water that was new since the last sampling (= younger than 3 to 4 days; $F_{new}$) in mobile soil water decreased from 7% at 10 cm depth to 4% at 20 cm depth, 3% at 40 cm depth, and 3% at 80 cm depth. Calculating $F_{new}$ for the 50% wettest sampling dates (based on total precipitation in the 3-4 days prior to sampling) reveals that mobile soil water contained more new water following wet antecedent conditions i.e., 10% at 10 cm depth, 6% at 20 cm depth, 4% at 40 cm depth, and 2% at 80 cm (Table 1). We also calculated the fraction of water that was younger than three weeks in mobile soil
water, that decreased from 51% at 10 cm depth to 31% at 20 cm depth, 25% at 40 cm depth, and 4 % at 80 cm depth for all sampling dates, and from 61% at 10 cm depth to 38% at 20 cm depth, 33% at 40 cm depth, and 2% at 80 cm depth for the wettest 50% of sampling dates (Fig. 4). Thus, while approximately two thirds of mobile soil water at 10 to 40 cm depths were typically younger than two to three months (as indicated by $F_{yw}$ – Fig. 3), relatively little of this water originated from the most recent precipitation (i.e., less than 3-4 days ago).

The $F_{new}$ in bulk soil waters (reflecting the fraction of water younger than 3 weeks, because this is the sampling interval of bulk soils) decreased from 34% at 10 cm depth to 25% at 20 cm depth, 21% at 40 cm depth, and 12% at 80 cm depth; for the 50% wettest sampling dates, the corresponding three-week $F_{new}$ values are 37% at 10 cm depth, 33% at 20 cm depth, 29% at 40 cm depth, and 9% at 80 cm depth (grey bars in Fig. 4). Thus, wet antecedent conditions increased the (three-week)
fraction of new precipitation found in bulk soil waters.

The $F_{new}$ in deep mobile water at 2 to 6 m depths (reflecting the fraction of water younger than 2 weeks, because this is the sampling interval in the deep boreholes) was very small (1.2%, and for the 50% wettest sampling dates, 1.5%). $F_{new}$ in discharge (sampled every day) was calculated for three-weekly, two-weekly, weekly, 3-day and daily aggregation intervals.
Across all sampling dates, 11% of streamflow was younger than three weeks, 12% of streamflow was younger than two weeks, 7% was younger than one week, 6% was younger than three days, and 3% was younger than a day. For the 50% wettest sampling dates, 12% of streamflow was younger than three weeks, 16% of streamflow was younger than two weeks, 13% was younger than one week, 12% was younger than three days, and 7% was younger than a day (Table 1).





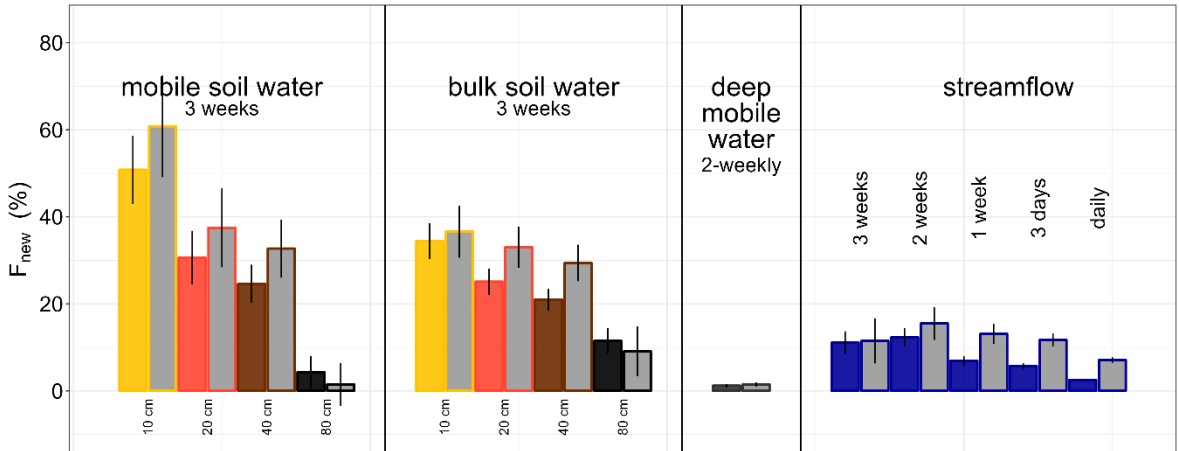

**Figure 4: New water fractions ($F_{new}$) in mobile and bulk soil waters (younger than three weeks) in 10, 20, 40 and 80 cm depth, deep mobile soil water sampled from boreholes (younger than two weeks) and streamflow (for 3-week, 2-week, weekly, 3-day and daily time-step aggregations) for all sampling dates (in colours) and the 50% wettest sampling dates (in shades). $F_{new}$ are typically smaller at greater depths, and larger following higher precipitation.**

**Table 1: Fractions of waters younger than 2-3 months ($F_{yw}$) and $F_{new}$ for different time-step aggregations for all sampling dates (in bold) and the wettest 50% of sampling dates (in italics) including the respective standard errors.**

|  |  | $F_{yw}$ [%] | $F_{new}$ [%] for all *and the wettest sampling dates* |  |  |  |  |
|---|---|---|---|---|---|---|---|
|  |  | 2-3 months | 3 weeks | 2 weeks | 1 week | 0.5 weeks | 1 day |
| **Mobile Soil Waters** | **10 cm** | 63 | **50.8 (±7.8)** *60.8 (±11.7)* | **33.4 (±5.1)** *56.5 (±8.7)* | **14.0 (±1.9)** *25.9 (±3.7)* | **7.2 (±1.1)** *9.7 (±1.9)* | - |
|  | **20 cm** | 68 | **30.6 (±6.2)** *37.5 (±9.1)* | **18.6 (±3.7)** *25.0 (±5.9)* | **11.3 (±2.0)** *18.4 (±3.8)* | **4.4 (±1.0)** *5.9 (±1.7)* | - |
|  | **40 cm** | 63 | **24.6 (±4.4)** *32.7 (±6.7)* | **12.3 (±2.5)** *20.0 (±4.5)* | **4.3 (±1.1)** *7.8 (±2.0)* | **3.1 (±0.6)** *4.4 (±1.2)* | - |
|  | **80 cm** | 26 | **4.3 (±3.7)** *1.5 (±4.9)* | **3.2 (±2.7)** *3.5 (±6.7)* | **2.9 (±1.5)** *2.1 (±2.9)* | **2.8 (±0.6)** *1.8 (±0.8)* | - |




| | | | | | | | |
|---|---|---|---|---|---|---|---|
| **Bulk Soil Waters** | **10 cm** | **66** | **34.4 (±4.1)** *36.6 (±6.0)* | - | - | - | - |
| | **20 cm** | **49** | **25.1 (±3.0)** *33.0 (±4.7)* | - | - | - | - |
| | **40 cm** | **36** | **21.0 (±2.5)** *29.4 (±4.2)* | - | - | - | - |
| | **80 cm** | **18** | **11.5 (±2.9)** *9.1 (±5.7)* | - | - | - | - |
| **Deep Mobile Waters** | **2-6 m** | **6** | - | **1.2 (±0.4)** *1.5 (0.5)* | - | - | - |
| **Streamflow** | | **18** | **11.1 (±2.6)** *11.5 (±5.2)* | **12.3 (±2.1)** *15.5 (±3.8)* | **6.9 (±1.1)** *13.1 (±2.3)* | **5.7 (±0.7)** *11.7 (±1.5)* | **2.5 (±0.3)** *7.1 (±0.7)* |

Building upon the comparison of wetter periods versus all data, we calculated the fraction of new water for different ranges of 3-week precipitation amounts. Fig. 5 shows that $F_{new}$ in mobile and bulk soil waters as well as in streamflow increased with more precipitation in the month preceding the sampling. $F_{new}$ in mobile soil waters increased from 28% to 68% at 10 cm depth, from 18% to 53% at 20 cm depth, from 19% to 42% at 40 cm depth. $F_{new}$ in bulk soil waters increased from 31% to 36% at 10 cm depth, from 23% to 38% at 20 cm depth, from 10% to 40% at 40 cm depth, and from 11% to 15% in

streamflow. These results reiterate the importance of catchment wetness for water transport and percolation.





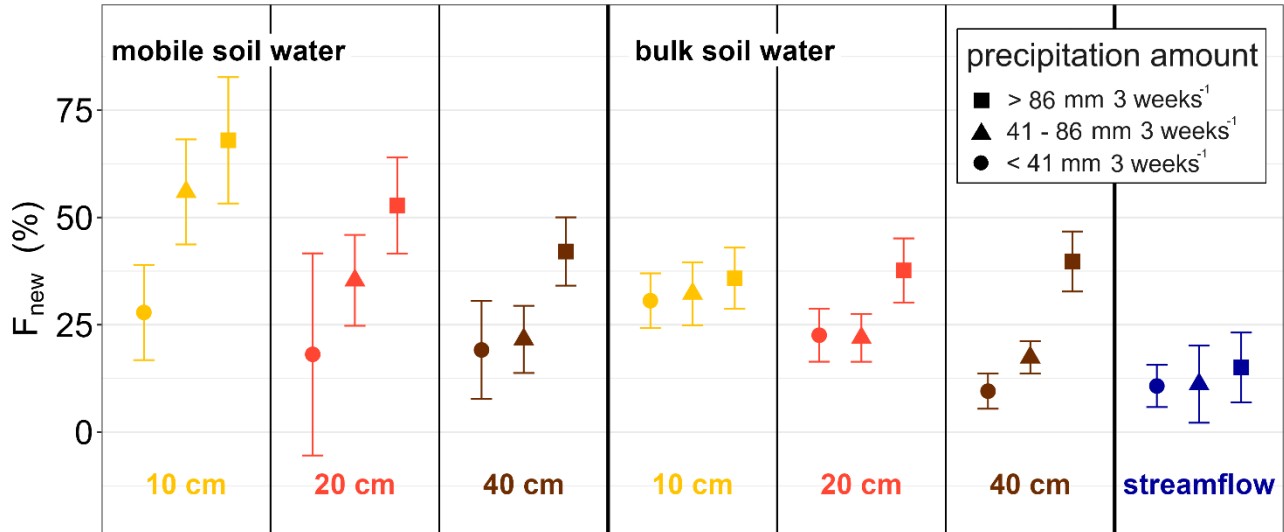

**Figure 5: New water fractions ($F_{new}$) in mobile and bulk soil waters at 10, 20 and 40 cm depth and streamflow as a function of precipitation totals during the 3-week period immediately preceding the sampling date (as indicated by different plotting symbols).**


### 3.4 Further interpretation of $F_{new}$ and $F_{yw}$ in soils and streamflow

The data suggest that soil new water fractions decrease with depth and increase with antecedent wetness, and that the deepest soil waters may be largely bypassed by new water flowing to streams. Approximately 18% of streamflow was younger than 2-3 months (as estimated from $F_{yw}$ – Fig. 2), 11% was younger than three weeks and 7% was younger than one week (both

estimated from $F_{new}$ – Fig. 4). Streams with similarly small fractions of young and new waters have been described elsewhere in the Alps (Floriancic *et al.*, 2023b), in an Andean floodplain (Burt et al., 2023), at Plynlimon in Wales (Knapp et al., 2019); such values are also common globally (Jasechko *et al.*, 2016). The effect of antecedent wetness on shorter transit times that we observed has also been quantified and observed in some of those studies (Floriancic *et al.*, 2023; Knapp *et al.*, 2019). While it may be intuitive that wet conditions allow for subsurface flow paths to connect pores, activate flow paths,

and transmit recent precipitation more quickly to streams, our measurements show that the same occurs vertically in soils (Fig. 3, 4, 5). Those patterns, and even the $F_{new}$ values in soils of 40 to 80 cm depth, were relatively consistent with those in streamflow, but not with the patterns and values observed in the deeper subsurface (i.e., deep mobile soil waters sampled from boreholes between 2 and 6 m depth). This indicates that streamwater was predominantly not derived from mobile soil waters below 2 m depth. Instead, waters stored in our hillslope below 2 m depth were typically much older than waters

draining from our hillslope (*sensu* age contrasts described in Berghuijs and Kirchner, 2017). That is to say that while the





isotopic signatures in soils between 10 and 80 cm suggest either translatory or well-mixed transport, flows from these soil layers to the stream may largely bypass deeper storages.

There are large isotopic differences between the mobile fraction of soil water (sampled by suction-cup lysimeters) and the entirety of soil water (represented by bulk soil water signatures). We found that bulk soil waters typically contained less young and new waters than mobile soil waters (Table 1). This is not surprising, as one would expect that mobile water is more easily replaced by recent precipitation as it is assumed to be less tightly bound in typically larger pores. While typically around two thirds of mobile soil waters at 10 to 40 cm depth were younger than 2-3 months (Fig. 3a), bulk soil waters at 20 to 80 cm depth were typically by-passed by recent precipitation and filled by waters predominantly originating from the winter season (Fig. 3b). At three- to four-day to three-week time scales, however (i.e., $F_{new}$ in mobile soil waters – Fig. 4, Table 1), we found that soil new water fractions decreased with depth, indicating that percolation from shallower to deeper layers typically requires more than three weeks.

Our results also suggest that soil water signatures and the fractions of young and new soil waters are significantly altered by evaporation and tree water uptake from specific pools. We hypothesize that forest trees, which preferentially access water in smaller pores (see discussion in Sprenger and Allen, 2020), cannot be emptying the bulk soil water stores, because bulk soils contain less than 50% "young" water (less than 2-3 months old) at all depths below 10 cm. Furthermore, the fact that bulk soil waters are systematically older than mobile soil waters at all depths below 10 cm implies that mobile soil waters must be largely bypassing the bulk soil water stores as they percolate through the profile. From previous studies, we found that tree water uptake at our site predominantly occurs around 40 cm depth (Floriancic *et al.*, 2023a; Martinetti *et al.*, 2023), although the small fractions of young (< 2-3 months) and new (< 3 weeks) bulk soil water in this layer ($F_{yw}$=36% and $F_{new}$=21%, respectively) indicate that any root water uptake is not primarily refilled by recent precipitation. By contrast, at 10 cm depth, the fractions of young water are large (> 60%) and consistent between mobile and bulk soil water, indicating that waters at 10 cm removed by evaporation, root water uptake, and percolation to deeper layers are substantially replenished by recent precipitation, with less bypassing than observed in deeper layers. Exact inferences about transport processes from these types of data are complicated by the localized uptake of water at different depths, which is often ignored in hillslope- or catchment-scale transport models.

A limitation to inferences made from comparing isotopes in bulk and mobile soil waters is that they are influenced by different uncertainties. While mobile soil waters were always sampled at the same location throughout the observation period, and thus reflect only the temporal variability of soil water isotopic signatures, bulk soil waters were sampled destructively at different locations in ~8 m$^2$ plots, thus reflecting both temporal and spatial variations in soil water isotopes. Another difference is that bulk soil water was extracted via cryogenic distillation, which introduces additional uncertainties (Orlowski *et al.*, 2016; Chen *et al.*, 2020). However, analyses such as the ones used in this study, which leverage variations



rather than absolute values, should not be sensitive to extraction artifacts if those artifacts bias all samples similarly (Allen and Kirchner, 2022).

### 3.5 Influences of vertical versus lateral flows along the hillslope

It is still unknown when or where lateral flow processes affect soil water, which is typically conceptualized and measured
from a vertically oriented perspective. To expand beyond this vertical perspective, we compared mobile and bulk soil waters from the two different sites at the top of the hillslope (the data shown in the previous figures) with those at a downslope streamside site (sampled for approximately one year from 04th April 2022 onwards). We found that the isotopic signatures in mobile soil waters were similar between the two upslope sites but significantly different from the downslope site (t-test, $p$-value < 0.05). However, the bulk soil water signatures were not significantly different between the three sites (t-test, $p$-value
> 0.05, Fig. 6).

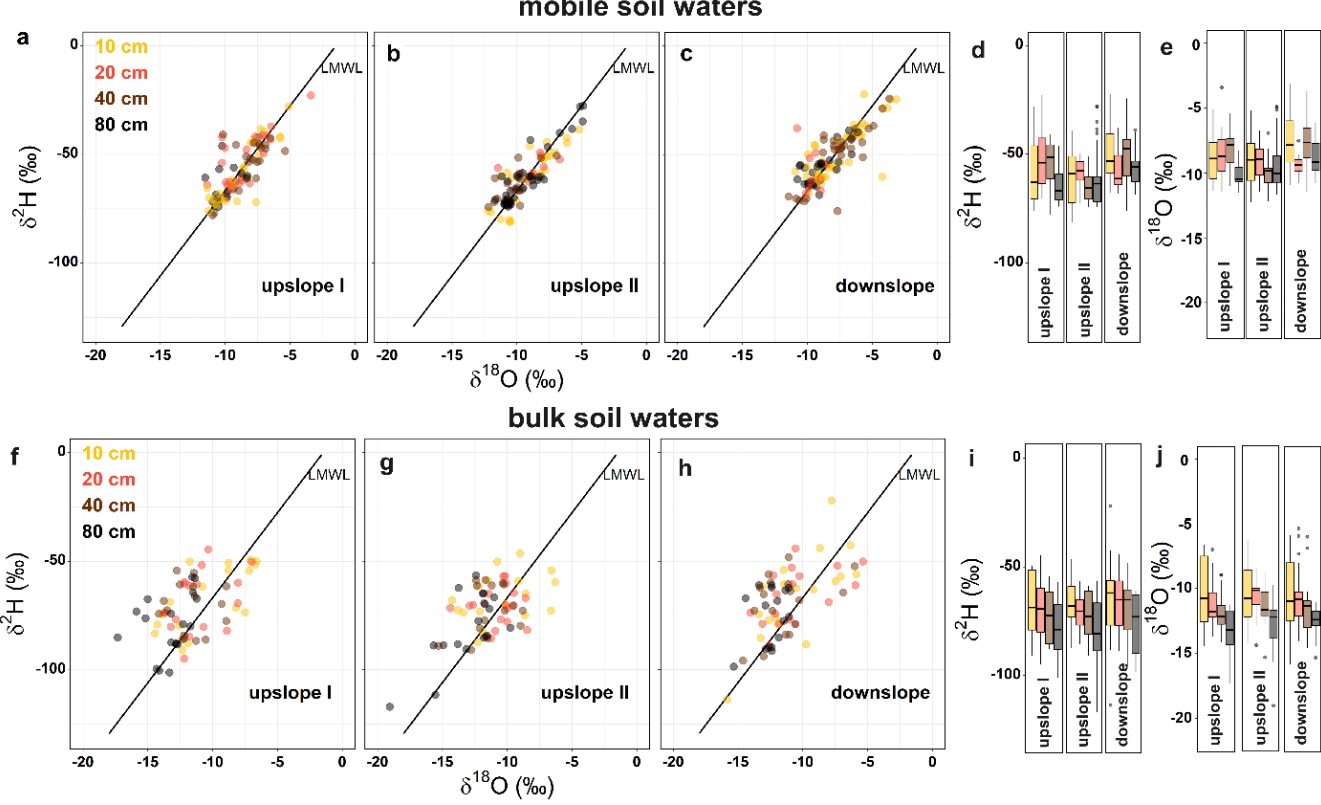

**Figure 6: Dual-isotope plots of mobile soil waters (upper panels) and bulk soil waters (lower panels) for the two sites on top of the hillslope (a, b, f, g) and the downslope site (c, h) including their respective boxplots (d, e, i, j). Mobile soil waters were different between the upslope sites and the downslope location, whereas bulk soil waters were quite**
**similar in all three sites. LMWL lines are calculated by reduced-major-axis regression (described in Harper, 2016)**



**because uncertainties in both $\delta^2H$ and $\delta^{18}O$ are equally important, whereas classic linear regression fitting assumes that the x-axis data have no error or uncertainty.**

We also compared the isotopic signatures in all 10 boreholes, of which five are located on top of the hillslope (upslope), three are in the middle of the hillslope (midslope) and two are in the saturated zone, 1.5 and 5 m from the creek (downslope). We found that seasonal isotopic variability was small for all three hillslope positions, and much smaller than the seasonal variability in streamflow (Fig. 7).



**Figure 7: Schematic diagram of the hillslope and borehole locations (a). Differences in the isotopic δ²H signatures of upslope and midslope deep mobile waters (b, c), downslope saturated groundwater (d), streamflow in the "Holderbach" creek (e) and precipitation (in grey in the background of the panels). The different colours indicate samples from different boreholes (five boreholes up-slope, four boreholes mid-slope and two boreholes downslope). The timeseries of the δ¹⁸O isotope signatures can be found in Fig. S3.**

Water samples extracted from the soils and sediments that were excavated when digging the boreholes were examined together with the overlying shallower bulk soil waters to extend the depth profile of the stored subsurface waters. In general, all bulk soil waters were isotopically lighter than average precipitation (the dashed line in Fig. 8) indicating that they are dominated by lighter winter precipitation. With increasing depth, winter precipitation became even more dominant in the bulk soil and sediment waters, indicating that less and less summer precipitation reaches deep layers of the subsurface at this site (Fig. 8). Although our site typically receives more precipitation during the summer half of the year (i.e., around 60% of annual precipitation), a much larger fraction of summer precipitation is lost through evaporation from the canopies, the forest floor litter layer and the upper soil layer (e.g., Gerrits and Savenije, 2011; Floriancic *et al.*, 2022). As a rough calculation, considering the interception loss of 20% of precipitation from forest canopy, another 18% from litter interception loss (see Floriancic *et al.*, 2022), roughly 18% quickly reaching streams (Fig. 2), and an unknown amount being withdrawn from shallow soils by trees, we suspect that less than one third of summer precipitation could end up recharging deeper storage. However, in winter, when these interception losses and evaporative demands are smaller, more winter precipitation is likely available to reach those depths, explaining the progressive decrease in isotope ratios with depth. An exception to this pattern is seen in Fig. 7, where the heavy-isotope signature of summer precipitation momentarily appears in the borehole records in October 2022, showing that these deep soil waters can respond to less-damped influxes, albeit with much more lag than seen in streamflow.



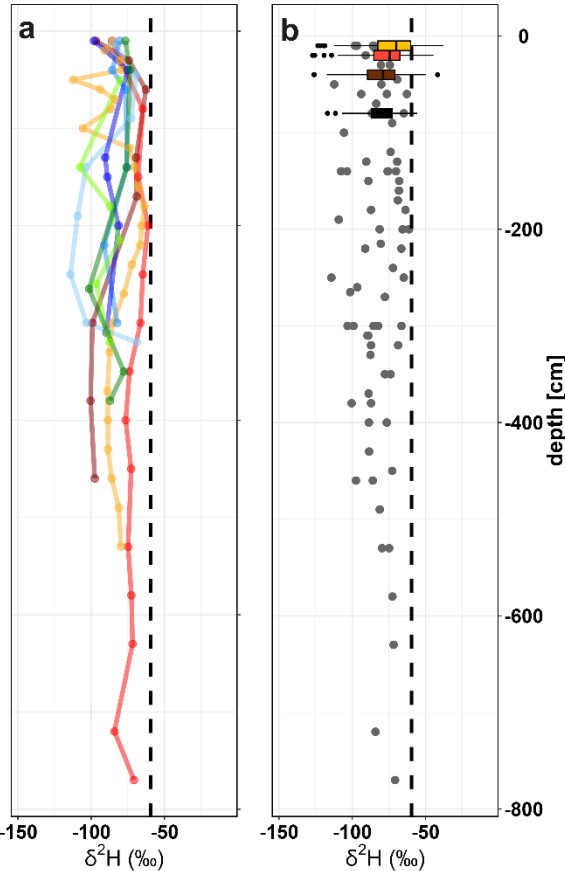

**Figure 8: Isotopic δ$^2$H signals in bulk soil waters during borehole drilling on November 22$^{nd}$ 2020 down to ~7 m depth (a), plotted also in grey with boxplots of bulk soil water δ$^2$H isotopic signatures for all regular bulk soil water samples across the three-year observation period for 10 cm in yellow, 20 cm in red, 40 cm in brown and 80 cm in black (b).**

**The dashed line indicates the mean precipitation δ$^2$H isotopic signature. Isotopic signatures in bulk soil water samples are typically lighter than the mean precipitation isotopic signatures, indicating a dominance of winter precipitation in bulk soil waters. The corresponding plots for δ$^{18}$O can be found in Fig. S4.**

### 3.6 Conceptualization of lateral and vertical hillslope water fluxes

Through interpreting Fig. 6 to 8 together, we identify two interesting trends. Bulk soil water signatures and variabilities were

similar among the different sites, both in value and pattern of progressive damping, indicating that similar vertical infiltration processes occur from 10 to 80 cm depth at each site. The mobile soil waters show some differences among sites, especially at the surface, shrinking at 40 to 80 cm and then disappearing at deeper depths (as seen in the borehole mobile waters). Thus, similar vertical transport processes predominate in the top 80 cm of both the upslope and downslope positions, with minor differences likely attributable to site-specific soil traits. However, somewhere below 80 cm there is a transition to deeper soil




waters being very well mixed (Fig. 7). From the data we cannot tell whether this results from vertical mixing or
homogenization due to lateral transport. However, any lateral transport or return flow from these deep, well-mixed pools has
no observable effect on the soils at 10 to 80 cm depth, even those located at topographically lower downslope sites.

Moreover, streamflow is much more isotopically variable than the deep mobile water samples, indicating that it is not
primarily generated by displacement of these deep mobile waters into the stream channel (Hewlett and Hibbert, 1967;
McDonnell, 1990). Even the boreholes in the saturated zone close to the creek (Fig. 7d) do not show the same seasonal
variation that is observed in streamflow. Likewise, contributions from streamwater do not control riparian groundwater
signatures, because the near-stream boreholes have vertical profiles similar to the upslope ones. The limited connection
between near-stream groundwater and streamwater is further indicated by the mismatch between shallower (10 to 80 cm)
and deeper mobile waters (2 to 6 m) during wet periods, as would be expected if riparian area groundwater levels rose to
generate streamflow. Instead, the seasonal isotopic cycle in streamflow indicates that streamflow is formed from a mixture of
deep subsurface waters, which exhibit nearly no seasonal isotopic cycle, and shallow (20-80 cm) soil waters, which exhibit a
much more pronounced seasonal cycle (Fig. 9). These shallow soil waters must primarily originate close to the stream
because we see no isotopic evidence for lateral dispersion or mixing of shallow soil waters across our hillslope sampling
sites.

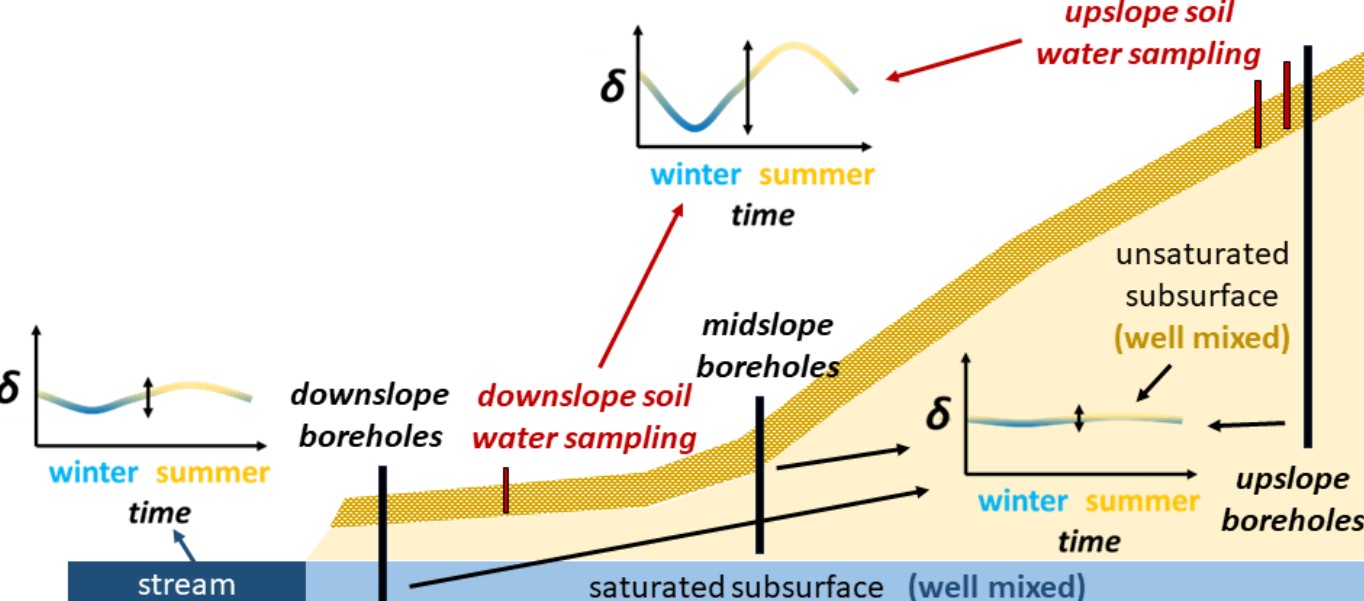

**Figure 9: Conceptualization of lateral and vertical hillslope water fluxes at the WaldLab Experimental Forest site.
The seasonal signal in soil water isotopes becomes weaker with depth. Below approximately 80 cm depth, soil waters
are well mixed with almost no seasonality. The seasonal isotopic cycle is larger in streamflow than in mobile waters**
**sampled from boreholes between 2 and 6 m depth, including boreholes that sample the saturated subsurface close to**





**the stream. This suggests that streamflow at our site is a mixture of waters from the shallow (20-80 cm) subsurface, which exhibit a pronounced seasonal isotopic cycle, and deeper layers, which exhibit almost no seasonal isotopic cycle.**

## 4 Conclusion

Interpreting mixtures of recent precipitation and older waters in the subsurface is key to understanding how, and how quickly, water is transported to its eventual fates as evapotranspiration and streamflow. By determining fractions of older waters versus recent precipitation in mobile soil water, bulk soil water, borehole mobile waters, and streamflow from continuous three-year records of isotope data across our forested hillslope transect, we have generated a novel suite of insights into hillslope water movement.


Our isotopic analyses demonstrate that fractions of young and new waters decreased with depth below the surface, but not monotonically. Roughly two-thirds of mobile soil waters at 10 to 40 cm depths were younger than two to three months, but less than 50% of bulk soil waters were similarly young at depths of 20 to 80 cm. Thus, most recent precipitation by-passed the smaller pores (represented by the bulk soil water samples) in the shallow layers. This isotopic evidence challenges

general conceptualizations of new precipitation inputs wetting dry soils or displacing previously stored waters from those soils. At our site this was only evident for the top 10 cm, which was also, unsurprisingly, strongly affected by evaporation. Streamwater was composed of 18 % precipitation younger than 2-3 months, 11 % younger than three weeks and 7 % younger than one week. These fractions of recent precipitation greatly exceeded those in deep subsurface waters at 2 to 6 m depth, even in boreholes situated immediately adjacent to the stream. The seasonal isotopic cycle in streamflow can only be

explained as a mixture of deep subsurface waters, with very little isotopic seasonality, and shallow soil waters, with more pronounced isotopic seasonality. The bulk soil waters and deeper mobile soil waters were dominated by light isotopic signatures reflecting winter precipitation, across both upslope and near-stream positions. Typically, fractions of recent precipitation in streamwater and soil water were higher under wet antecedent conditions, indicating accelerated transport through the hillslope hydrologic system.


These observations illustrate how measurements of isotopic variability across different subsurface depths, hillslope positions, and time scales can help to constrain potential flow processes delivering precipitation to deep soils and streams.

**Data availability**

The data that support the findings of this study are available from the corresponding author upon reasonable request.



**Author contribution**

MF designed and carried out the experiments and field sampling, processed the data and prepared the manuscript with contributions from all co-authors.

**Competing interests**

The authors declare that they have no conflict of interest.

**Acknowledgements**

The project was funded through Waldlabor Project Grant 4.20.P01 awarded to Marius Floriancic. We thank the Waldlabor Zürich (Martin Brüllhardt – www.waldlabor.ch) as well as the forest owner Jakob Heusser for their cooperation and support. Further    information    on    the    WaldLab    Forest    Experimental    Site    can    be    found    here:
https://hyd.ifu.ethz.ch/ecohydlab/waldlab.html. We acknowledge support during data collection and laboratory work from the technical staff and student helpers working at the WaldLab Forest Experimental Site (Lena Straumann, Anna Lena Könz, Maria Grundmann, Lara Virsik, Elisa Hage, Stefano Martinetti, Fabian Strittmatter, Adrian Kreiner, Mark Bühler, Gian-Luca Cavelti, Lucien Biolley and Martin Huber). We thank Tracy Napitupulu and Björn Studer for isotope analyses and Annika Ackermann, Roland A. Werner and Nina Buchmann for sharing the cryogenic extraction laboratory.

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
