# Peer review of "Young and new water fractions in soil and hillslope waters"

_EGUsphere, 2024_

## Author Response (AR1)

**We thank the editor & reviewers for the interesting comments and suggestions. Below we provide a detailed point-by-point response (in bold) to the editor & reviewers' comments (*in italic*).**
* * *
**Editor comments:**

*Dear authors,*

*the paper received two favorable reviews proposing a couple of changes and suggestions, but in general only demanded minor revisions. Please provide a revised version with the revisions, an document with all changes and detailed response letter to the suggestions of the reviewers.*

*Best regards*

*Markus Weiler*

**Thank you for the very positive feedback. We provide the necessary changes in the revised version of the manuscript and replied to the reviewer comments below.**
* * *
**Reviewer 1:**

*Overview: The authors presented a very interesting experimental design to estimate the water-ages of streamflow and soil-water. They gathered a high temporal resolution stable water isotope dataset over 3 years from different hydrological compartments like precipitation and streamflow as well as soil-water and ground water depth profiles. By fitting sinusoidal curves to the observed signatures and comparing the curves amplitude, they showed that the fraction of young and new water decreases with depth. They also showed that streamflow in their catchment was a mixture of deeper and more shallow soil water.*

*Overall, I found the paper pleasant to read and belief this is a great contribution to the field of catchment hydrology and travel-time assessment. The application of the developed method by Kirchner (2016a; 2016b) on such a high resolution stable water isotope dataset, especially with the focus on soil-water, is novel and will be a great addition to the current literature. I think the writing could be slightly improved. Please see comments below.*

**Thank you for the very positive feedback.**

*Line by line comments:*

*L 26: Leave out Switzerland and only focus on the global distribution of forests in your first introduction sentence*

**Thanks, we removed Switzerland.**

*Figure 1: Add elevation contour lines so reader gets a better sense of the plots topography*
**The covered area in the schematic diagram is roughly 30x10m, there is almost no gradient. We added this in the caption.**

*Figure 2a: Could change streamflow from point to line plot*
**Thank you, we updated the plot accordingly.**

*L 197: You introduced "piston flow" as "translatory flow"; decide on one term for consistency*
**Thanks, we chose piston flow to make the wording more consistent.**

*Figure 3: While a great colorblind color-palette, the point transparency makes it hard to distinguish between points in (a); maybe try solid points?*
**We experimented with this to try to increase readability, and now show the points without transparency.**

*Figure 3 Caption: "… in mobile (a) and bulk soil waters (b) of 10, 20, 40 and 80 cm depth and in deep mobile waters (c) collected in boreholes of 2 to 6 m depth."*
**We changed this as suggested.**

*Also add that the percentages below the depths indicate Fyw, otherwise the reference in Line 238 makes no sense*
**Thank you, we'll add this information in the figure caption.**

*Section 3.3, Figure 4 as well as Table 1 repeat a lot of information, especially information that is plotted in Figure 4 is already described in text, plot and table. Here I would advocate for less redundancy and maybe push the table back to the supplementary material.*
**Thank you for this suggestion. We see the repetition between Table 1 and Figure 4, but table 1 includes many more metrics as well. Thus, we prefer to keep it in the main text, figure and table.**

*Literature*
*Kirchner, J. W. (2016a). Aggregation in environmental systems – Part 1: Seasonal tracer cycles quantify young water fractions, but not mean transit times, in spatially heterogeneous catchments. Hydrology and Earth System Sciences, 20(1), 279–297.* *https://doi.org/10.5194/hess-20-279-2016*

*Kirchner, J. W. (2016b). Aggregation in environmental systems – Part 2: Catchment mean transit times and young water fractions under hydrologic nonstationarity. Hydrology and Earth System Sciences, 20(1), 299–328.* *https://doi.org/10.5194/hess-20-299-2016*

**Reviewer 2:**

*General comments:*
*The authors present an impressive isotope dataset collected from precipitation, different reservoirs of a hillslope's soil water and the adjacent stream. They use these data to derive the young and new water fractions of soil and stream water. Such a detailed investigation is certainly of interest for the hydrological community and I recommend publication after moderate revision.*
**Thank you for the very positive feedback.**

*Formally, the authors decided to combine the result and discussion section which unfortunately tempted them to neglect a proper description of some of the results. This made recalculation of the presented young and new water fractions impossible. The authors should at least present the equations of the fitted sinusoidal cycles prior to discussing their amplitudes and phase shifts.*
**Thank you. We understand why readers may want this and added the parameters of the fitted sinusoidal cycles in the supplement of the revised version of the manuscript.**

*Further, I have issues with some of the presented isotope data. There appears to be a notable shift in bulk water isotope data which potentially must be attributed to extraction artefacts. While this may not necessarily affect the interpretation of sinusoidal amplitudes and phase shifts, I feel that the drawn conclusions pointing at certain seasons of replenishment need to be discussed again.*
**We updated the discussion to include text that addresses the uncertainties in bulk soil water samples (outlined below in the specific comments). A directional bias introduced by extraction should not impact amplitude or phase terms, and thus the interpretation itself remains robust to any such artefacts.**

*Also, I am unhappy with the authors' definition of the mobile water phase. It was obtained with 0.7 bar applied suction. This translates to a pF value of 2.8 which exceeds the commonly accepted pF 1.8-2.1 for threshold between mobile and immobile water. Additionally, the manuscript seems to suggest that mobile and bulk water are two separate reservoirs and not mobile water being part of the bulk water reservoir.*
**This is an important remark. We see how our language could be interpreted to suggest that separation. Of course, mobile soil waters are also part of bulk soil water and not two separate entities. We thoroughly edited the revised manuscript to improve this. There was a typo in the manuscript, we applied a suction of 0.6 bar NOT 0.7 bar. Also, we made it more clear that for our data collection, the term "mobile soil water" refers to water extracted by a tension of < 0.6 bar.**

*I provide a list of specific comments below.*

***Specific comments:***

*L14: "not surprisingly" seems to contradict the "poorly understood" from L11.*
**We will delete the 'not surprisingly' because it is unnecessary.**

*Figure 1: What is the slope and exact position of this site? Can it be considered representative of the entire stream catchment regarding hydrological response?*
**The covered area in the schematic diagram is roughly 30x10m, there is almost no gradient. It is unlikely that such a small area would be representative of the entire stream catchment (either here or at any other site), and our analysis does not require that. We will provide a more detailed map of the entire catchment in the supplement of the revised manuscript.**

*L135: 0.7 bar applied suction translates to pF 2.8. Why was such a high suction applied and why was it the same for all depths regardless of the different resulting water columns assuming that all sample collectors were placed on surface level?*
**This is an astute and relevant remark, and we appreciate the reviewer's attention to this detail. Ultimately there are always choices to be made in sampling strategies and these introduce tradeoffs. Here, we opted for consistency and repeatability in our sampling strategy. While there are ways in which this is not optimal, we expect it to be a protocol that is most likely to be maintained into the future. Also, unfortunately there was a typo in the original manuscript, we applied a suction of 0.6 bar NOT 0.7 bar.**

*L137: How much soil was sampled for extraction, i.e. from what depth ranges was soil obtained using the auger? Did that ranges match the lengths of the suction cup tips?*
**Yes, the ranges of bulk soil sampling matched the lengths of suction cups. We now provide a more detailed description in the revised version.**

*L143ff: What were the elevations of the boreholes or rather the observed groundwater tables relative to the adjacent stream water level? Did you observe a gradient and (how) did it change over time?*
**We provide a table with borehole elevation in the supplement of the revised manuscript. Groundwater tables only exist in two of the boreholes. The exact evaluation of groundwater table relative to streamflow would require a very precise measurement of sensor location that is unfortunately currently not available.**

*L154: Why was 95% recovery rate considered sufficient? Araguas-Araguas et al. (1995, doi.org/10.1016/0022-1694(94)02636-P) demonstrated that a recovery rate of better*

*than 98% is necessary to obtain accurate isotope data with their accepted measurement uncertainty being higher than is this study. Isotope values flawed by incomplete extraction tend to be lower thus erroneously pointing at water from colder seasons.*

**Thank you, we now understand that the description provided was not sufficient. In the reference provided (Berhard et al. 2024) we showed that for 89% of samples the extraction efficiency was > 98% and for the remaining 11% of the samples, the extraction efficiency was between 96% and 98%. We updated the description accordingly.**

*L157: You report the precision of the analyser being derived from measurement replicates. What was the accuracy of your method, i.e. the deviation from target values?*

**We report the precision derived from measurement replicates of standards.**

*L167: How much is "much"? Without numbers ideally presented together with the equations of all fitted sinusoidal cycles this statement cannot be quantitatively retraced.*

**Thank you. We added the amplitudes (and other fitting parameters) of the fitted sinusoidal cycles in the supplement (Table S2) of the revised version of the manuscript and refer to this table in the main manuscript.**

*L171: Why "median" and not (depth-weighted?) mean? Further, the difference between the reported numbers is only a fraction of the reported measurement precision. Why do you bother discussing such small differences?*

**The median is useful as a measure of central tendency (implicitly time-weighted) that is not sensitive to outliers. This is of value to readers. The measurement precision is 0.2 for $\delta^{18}O$, but we mistakenly have a typo referring to $\delta^2H$ here.**

*L175: Also here, "ratio" is lacking numbers.*

**Thank you. We added the amplitudes (and other fitting parameters) of the fitted sinusoidal cycles in the supplement of the revised version of the manuscript (Table S2) and refer to this table in the main manuscript.**

*Figure 2: Does relative streamflow refer to runoff (in volume unit per time unit)?*

**This is a relative scale, normalized by the minimum and maximum between 0 and 1.**

*L184: Please move the description of how sinusoidal cycles were fitted to the obtained isotope data to the method section. Please also include a statement of the degree of freedom you allowed. I assume, the frequency was pre-set to 365 days and only amplitude and phase shift were fitted?*

**This is correct. We'll update the description in the methods section.**

*L186ff (and other figure captions): I feel that these interpretations and descriptions unnecessarily inflate the figure caption. The respective statements already appear in the main text.*

**This is a stylistic choice, and we believe that a one-sentence summary of the main message of the figure is essential for readers, especially as it is very often hard to find the relevant information in bulk text that is (depending on typesetting) sometimes even on different pages. In our experience, many readers will simply not get the main information by just browsing through the paper, if the figure caption does not contain a short, one-sentence summary. Thus, we feel that such summaries help to make manuscripts easier to understand.**

*L196: Statements regarding phase shifts are only meaningful when the underlying numbers are available.*

**Thank you. We now provide the detailed description of the sinusoidal fits including the phase shifts in the supplement of the revised version of the manuscript (Table s2).**

*L205: To me, the isotope values of bulk water being lighter seems to be an artefact resulting from incomplete extraction (see also comments on Figure 6). Therefore, I would be careful drawing the conclusion regarding winter precipitation.*

**As outlined before, incomplete extraction is not a major concern leading to systematic shifts in our data, and we will update the description in the methods section as outlined above. There are multiple more physically-based reasons to believe that the predominance of winter waters in bulk soil (and xylem) waters is not (solely) related to extraction artefacts (as discussed in section 3.4) but to the limited input of summer precipitation to forest soils due to substantial interception losses, at least at our site (see also Floriancic et al. 2024).**

*L208f: At some time of a year winter precipitation is actually the recent precipitation. Why would bypass only occur in summer?*

**We updated the discussion to make this line of argumentation clearer to readers: Bypass flow is inferred to be dominant, whenever pores are already filled with water, independent of the season. However, conceptually speaking, pores are emptied throughout the summer seasons with the driest soils observed during the end of the growing season (in autumn). Overall, only a little summer precipitation makes it to the soil (due to interception and evaporation, section 3.4 and Floriancic et al. 2024) Therefore deeper pores are predominately filled by whatever is available after they are emptied, which in our case is mostly winter precipitation.**

*L232f: Do you happen to have soil water content data (e.g. via relative post-extraction soil sample weight losses) and (how) would they make your claim more plausible?*

Unfortunately, we do not understand this remark about making our claim more plausible. Moreover, the sample sizes of soils collected for extraction is quite small and unlikely to yield representative soil water contents.

*L250 (and L252): Isn't it remarkable that the 3-week fraction which INCLUDES the 2-weeks fraction is actually smaller than the latter alone?*
We appreciate the reviewer's careful attention to details because it benefits our paper. However, uncertainties (i.e., the standard errors) of these estimates (given in Table 1) must be considered, and 11.1±2.6 is not smaller than 12.3±2.1, and 11.5±5.2 is not smaller than 15.5±3.8. We will add text to point out that it may seem surprising that the 3-week fraction, which includes the 2-weeks fraction, might be smaller than the 2-week fraction, but consideration of uncertainties would imply that these values are indistinguishable.

*L261: Are the uncertainties derived from the scatter of isotope data, i.e. the deviations from the sinusoidal cycles?*
No the uncertainties are standard errors, derived from the ensemble hydrograph separation calculations for new water fractions, this is not done by fitting sinusoidal cycles.

*L275: Why did you exclude data from 80 cm from this figure?*
Unfortunately, we only started sampling bulk soil waters at 80cm depth 14 months later, resulting in a much shorter timeseries, thus splitting the data was not reliable. We will explain this in the text.

*L283: Is it Floriancic et al., 2023 a or b?*
Thank you, it is Floriancic et al. 2023 b.

*L294: Please provide the numbers this statement is based on.*
We here refer to previous results shown in Figure 3 and described in the text related to Figure 3; we added a reference to the previous section.

*L287: Delete "assumed to be"*
L 297? Thank you, we will remove it in the revised version of the manuscript.

*L305 (and elsewhere): By "evaporation", do you mean entire loss a certain fraction (with no effect on isotopic composition and deuterium excess on the remaining reservoir) or partial loss from the total reservoir (resulting in the aforementioned effects)?*
Thank you. We understand that this is not clear. We refer to water intercepted and evaporated in entirety, thus not reaching the soil at all.

*L307ff: This statement seems to suggest that mobile water and bulk water are two separate reservoirs. But isn't mobile water part of bulk water? The extraction process will certainly drive out not only the more tightly bound water, right? Assuming that one is part of the other, can you make a statement regarding the relative proportions under wet vs. dry antecedent conditions?*

**Thank you. We updated this section to make it clear to readers that although bulk soil waters also contain mobile soil waters, the relative fraction of the two likely depends on soil wetness. As soils dry, the isotopic signal in bulk soil waters is presumably increasingly dominated by the waters stored in small pores that are not part of the mobile soil water pool.**

*L325: These prerequisites ("all samples have the same bias") seem to be contradicted by Figure 6 where data scatter more in the case of bulk soil water isotope data. In Figure 6 it also appears to me that not only are isotope data shifted towards the lower left but also the centers of gravity seem to be shifted at a slope < 8 above the LMWL. If so, this would indicate artefacts resulting from incomplete extraction. While that may not so much flaw interpretations of sinusoidal amplitudes and phase shifts, it certainly makes interpretations of the season of bulk soil water replenishment questionable.*

**The statement "all samples have the same bias" does not exist in the submitted manuscript. In line 325ff we discuss extraction artefacts in bulk soil water samples *"However, analyses such as the ones used in this study, which leverage variations rather than absolute values, should not be sensitive to extraction artifacts if those artifacts bias all samples similarly"*. This means that an extraction artifact that shifts all samples of a particular type (in this case, bulk soil) by similar average amounts (i.e., bias) will not affect the inferred young water fractions or new water fractions (see Kirchner 2016a and Kirchner 2019). We also explain that bulk soil waters also contain spatial (not only temporal) variability in lines 322ff *"While mobile soil waters were always sampled at the same location throughout the observation period, and thus reflect only the temporal variability of soil water isotopic signatures, bulk soil waters were sampled destructively at different locations in ~8 m2 plots, thus reflecting both temporal and spatial variations in soil water isotopes."* From this short comment, we do not follow the logic of how spatial variability and its interplay with extraction artefacts would lead to a seasonal bias in bulk soil water samples.**

*L332ff: Is the significance and non-significance a result of the differences in scatter?*

**We used t-tests to assess significance, which takes scatter into account. Certainly, greater scatter makes it less likely to find significant differences. However, in the case of findings of significant differences, there would still need to be differences in the mean values and so scatter alone could not explain significance.**

*Figure 6. Would it be possible to also present precipitation and stream water data in dual isotope space?*

**We added a dual isotope plot of precipitation and streamflow in the supplement of the revised version of the manuscript.**

*L344: Please move the site description to the method section. Please provide the underlying numbers before discussing the differences in variation.*

**We now explicitly refer to Figure 6 where we show the data in dual isotope space and boxplots. We explicitly mention this additional explanation here, because we thought readers will need this information directly where they see the results. Thus, we'd like to keep it here.**

*L355: Please move this sentence to the method section.*

**We think that this sentence is important here, as it serves to remind readers of the context of these samples and how they differ from the others discussed.**
* * *
**References:**

**Bernhard F, Floriancic MG, Treydte K, Gessler A, Kirchner JW and Meusburger K; (2024) Tree- and stand-scale variability of xylem water stable isotope signatures in mature beech, oak and spruce; Ecohydrology; https://doi.org/10.1002/eco.2614**

**Floriancic MG, Allen S and Kirchner JW; (2024) Isotopic evidence for seasonal water sources in tree xylem and forest soils; Ecohydrology; https://doi.org/10.1002/eco.2641**